# Resolving the fine structure in the energy landscapes of repeat proteins

## Research Article

Protein folding; Molecular dynamics; Folding funnel; Energy landscape visualisation

**Author for correspondence:**
*Diego U. Ferreiro,
E-mail: ferreiro@qb.fcen.uba.ar
*Vitor B.P. Leite, E-mail: vitor.leite@unesp.br

M.N.S. and R.G.P. equally contributed to this work.

Murilo N. Sanches[1] , R. Gonzalo Parra[2] , Rafael G. Viegas[1,3] ,
Antonio B. Oliveira Jr.[4] , Peter G. Wolynes[4] , Diego U. Ferreiro[5]* and
Vitor B.P. Leite[1]*

[1]Department of Physics, Institute of Biosciences, Humanities and Exact Sciences, São Paulo State University (UNESP), São José do Rio Preto, SP 15054-000, Brazil; [2]Barcelona Supercomputing Center (BSC), Barcelona, Spain; [3]Federal Institute of Education, Science and Technology of São Paulo (IFSP), Catanduva, SP 15.808-305, Brazil; [4]Center for Theoretical Biological Physics, Rice University, Houston, TX, USA and [5]Instituto de Química Biológica de la Facultad de Ciencias Exactas y Naturales, C1428EGA Buenos Aires, Argentina

## Abstract

Ankyrin (ANK) repeat proteins are coded by tandem occurrences of patterns with around 33 amino acids. They often mediate protein–protein interactions in a diversity of biological systems. These proteins have an elongated non-globular shape and often display complex folding mechanisms. This work investigates the energy landscape of representative proteins of this class made up of 3, 4 and 6 ANK repeats using the energy-landscape visualisation method (ELViM). By combining biased and unbiased coarse-grained molecular dynamics AWSEM simulations that sample conformations along the folding trajectories with the ELViM structure-based phase space, one finds a three-dimensional representation of the globally funnelled energy surface. In this representation, it is possible to delineate distinct folding pathways. We show that ELViMs can project, in a natural way, the intricacies of the highly dimensional energy landscapes encoded by the highly symmetric ankyrin repeat proteins into useful low-dimensional representations. These projections can discriminate between multiplicities of specific parallel folding mechanisms that otherwise can be hidden in oversimplified depictions.

## Introduction

The structural domains that are visible in X-ray crystal structures of proteins are often thought of as modules that can fold, function and evolve independently. Nevertheless, large proteins made up of tandem repetitions of apparently modular structure do not fold by independently organizing those modules, but rather the modules cooperate in stabilizing structural intermediates that comprise several repeat units (Paladin et al., 2020). Repeat proteins can be classified into many different categories, based on the length of their repeating units, the type of secondary structure elements of which they are composed and their overall architecture. A specific class of repeat proteins, the so-called solenoids, is constructed from 20 to 40 similar amino acid stretches that fold up into elongated architectures of stacked repeating structural motifs. In Fig. 1, we show three examples of alpha solenoids, from the ankyrin (ANK) repeat protein family. Proteins in this family are made up of a variable number of repetitions of a 33-residue-length structural motif (Parra et al., 2015). For these proteins, exactly how to separate the structure into 'domains' is not obvious in a mechanistically correct way (Parra et al., 2013; Espada et al., 2015). A coarse representation of ankyrin repeat proteins as quasi-1D objects has, however, yielded surprisingly rich insights into their folding dynamics (Petersen and Barrick, 2021). The one-dimensional representation of the stabilization mechanism of this class of repeat proteins arises because they are stabilized only by interactions within each repeat and between neighbouring repeats, there being no obvious contacts between residues much more distant in sequence. While ankyrin repeat proteins can be pictured as elongated objects that can be broken down to repeat units, the precise mechanism of the folding of the array reflects subtle balances and imbalances between the energetics within the repeats and the interaction between repeats (Ferreiro et al., 2008). For many natural repeat proteins, it has been shown that weakening the energetic links between repeats leads to the breakdown of cooperativity and the appearance of folding subdomains within an apparently regular repeat array (Aksel et al., 2011). In general, the folding mechanisms are defined by an initial nucleation in some region of the repeating array and the propagation of structure to their near neighbours. When the local energetics are similar along the assemblage, parallel folding routes can be identified (Werbeck and Itzhaki, 2007; Aksel and Barrick, 2014) and the routes can be switched by (de)stabilising regions along the array (Tripp and Barrick, 2008; Werbeck et al., 2008). Thus, the energy landscapes of repeat proteins can be very rich and amenable to design (Galpern et al., 2022). Most importantly, in various cases, the detailed folding

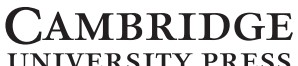

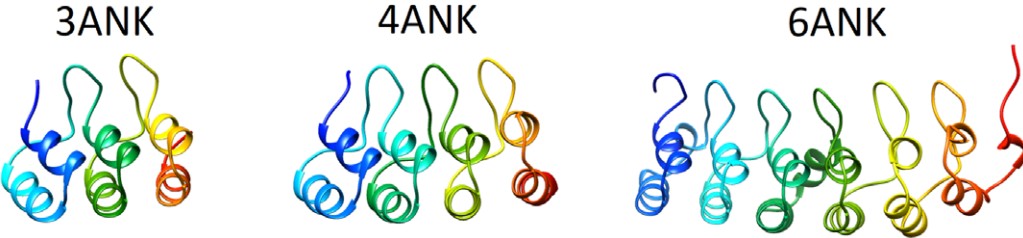

**Fig. 1.** Representation of the tertiary structures of Ankyrin repeat proteins. The high-resolution structures of the proteins studied are shown in ribbons coloured in blue to red from *N* to *C* termini. 3ANK corresponds to PDB 1N0QA, 4ANK with PDB 1N0R and 6ANK with PDB 1NFIE.

mechanism of repeat arrays has been identified to play a major role in their biological function (Löw et al., 2008; Barrick, 2009; Ferreiro and Komives, 2010; Kumar and Balbach, 2021). How can we picture the energy landscapes of these systems?

The notion of protein folding Energy Landscapes has illuminated many of the mysteries of how proteins fold. Based on solid theoretical arguments, the synergy of experiments and simulations has shown that the folding of globular proteins occurs on landscapes that can be described as rough funnels (Wolynes *et al.,* 2012), where both the solvent averaged free energy of configurations and their entropy decrease nearly in parallel as structures ever more closely approximate the native state. This correlation arises from the 'Principle of Minimal Frustration' (Bryngelson and Wolynes, 1987), which argues that the native interactions of evolved proteins are on average stronger than competing possible non-native interactions. The fact that simple structure-based models of protein folding can recapitulate even detailed features of protein folding mechanisms, such as $\varphi$-values, further supports this view (Clementi, 2008). Still, both topological frustration and energetic frustration that accompany functional constraints, such as binding and catalysis, play roles in the folding mechanism of several globular protein systems (Ferreiro *et al.*, 2018).

It seems the energy landscapes of repeat proteins follow the funnelling criterion, both overall (Mello and Barrick, 2004; Ferreiro *et al.,* 2008) and in the folding of consecutive units and individual units, thus suggesting a landscape of funnels within funnels (Ferreiro and Komives, 2007). The fact that repeat proteins can be treated as quasi-one-dimensional objects, however, weakens the necessity for a deeply funnelled landscape, as conflicting interactions that may arise from the frustration of interactions far distant in sequence are not as predominant as in globular domains. Simple structure-based models of repeat-protein folding have been shown to predict folding behaviour consistent with the overall behaviour of repeat arrays seen in the laboratory (Ferreiro *et al.*, 2005; Barrick *et al.*, 2008). More complex models, such as all-atom simulations, have been applied to study the folding of ankyrin repeat proteins under force (Serquera *et al.,* 2010). In these, parallel routes, intermediates, and partial folding of repeats have been described and characterised, and these may be stabilised by non-native interactions. Moreover, high-temperature unfolding of designed ankyrin repeat proteins has been analysed and used to redesign the terminal repeats (Interlandi *et al.,* 2008). Since we want to visualise a large fraction of the energy landscape, we use here the coarse-grained AWSEM model (Davtyan *et al.,* 2012), and we make use of a novel energy-landscape visualisation method (ELViM; Oliveira *et al.,* 2019) to analyse the folding of repeat arrays in the ankyrin family.

Given a dataset of sampled conformations classified using an appropriate metric, ELViM first calculates a matrix comprised of the internal distances between every pair of sampled conformations. Ideally, these distances should correlate with energetic changes on the landscape. This matrix, which represents the dataset in the high-dimensional phase space, is then projected onto an effective 2D or 3D phase space, which preserves in an optimal way these distances. Since folding is most facile between similar structures, ELViM provides a meaningful visualisation of the folding mechanism. In this article, we apply ELViM to explore the energy landscapes of three Ankyrin Repeat Proteins having each 3, 4 and 6 repeats (Fig. 1) and document ELViM's ability to capture the dynamic energetic behaviour of the different repeats and the interactions among them as the protein navigates its conformational space. ELViM by faithfully visualising the vast information in the content of molecular simulations offers a powerful tool to study repeat-protein folding mechanisms.

## Methods

### Simulations details

We performed coarse-grained molecular dynamics simulations using the AWSEM-MD suite (Davtyan *et al.*, 2012). We used a structure-based model called AMH-Go that includes a nonadditivity term that allows for a more realistic simulation of the cooperativity among native interactions (Eastwood and Wolynes, 2001). By virtue of being a structure-based model, this landscape is perfectly funnelled. The measure of similarity between structures *k* and *l* used here is based on internal distances between amino acids, and it is given by

$$q_{k,l} = \frac{1}{N_p} \sum_{i,j \in \text{pairs}} \exp\left[ \frac{-\left(r_{i,j}^k - r_{i,j}^l\right)^2}{2\sigma_{i,j}^2} \right], \qquad (1)$$

where $N_p$ is the total number of pairs of residues, $r_{i,j}^k$ ($r_{i,j}^l$) is the distance between the residues *i* and *j* in the conformation $k(l)$ and $\sigma_{i,j}$ is the Gaussian standard deviation and it accounts for the increasing variances when one considers residues far from each other along the primary sequence. It is defined as $\sigma_{i,j} = \sigma_0 |i - j|^\varepsilon$, with $\sigma_0 = 1$ Å and $\varepsilon = 0.15$, as used in previous studies (Lätzer *et al.,* 2007). $q_{k,l}$ is normalised and unitless. The energy landscape will locally correlate with this relative *q* distance measure. The particular similarity between a structure *k* and the native state *n* of the entire protein, $q_{k,n}$, can be used as an approximate reaction coordinate, and it is referred to as the global reaction coordinate or $Q_w$. The sum in Eq. (1) when carried out over a small subset of

residues, such as a single repeat, defines a local coordinate $Q_W^{Rn}$, and when the sum is associated with only a single residue $i$, $Q_W^i$. Another coordinate that is well established and often used as a reaction coordinate is the fraction of native contacts, $Q_o$ (Best *et al.*, 2013), defined as

$$Q_o(X) = \frac{1}{N_s} \sum_{i,j \in S} \frac{1}{1 + \exp\left[\beta^0 \left(r_{ij}(X) - \lambda r_{ij}^0\right)\right]}, \quad (2)$$

where $r_{ij}(X)$ is the distance between the residues $(i, j)$ in a conformation $X$, $r_{ij}^0$ is the distance between the amino acid in the corresponding pair in the native state, $S$ is the set of all pairs of native contacts $(i, j)$ from the native structure, $N_S$ is the number of pairs in $S$, $\beta^0$ is a smoothing parameter and $\lambda$ is a factor that takes into account the fluctuations of the contacts. In this work, we consider any pair of atoms as being in contact when the partners are more than three residues apart along the chain and if $r_{ij}^0 < 1.2$ nm. The parameters in Eq. (2) were taken to be $\beta^0 = 50$ nm$^{-1}$ and $\lambda = 1.2$ nm, as suggested for AWSEM model (Habibi *et al.*, 2016).

To estimate the folding temperature $T_F$, we performed melting simulations heating the systems from a low temperature (300 K) to sufficiently high temperatures to ensure unfolding (800 K). The temperature variations were made along 10 million steps, with a time step of 3 fs. During this process, due to the cooperativity among native interactions, a sharp transition occurs in the reaction coordinate $Q_W$ as a function of temperature, where the system goes from largely folded to largely unfolded states. $T_F$ is estimated to be near the midpoint of the transition between these states.

We then performed constant temperature simulations at the estimated $T_F$ using the Umbrella Sampling method (Torrie and Valleau, 1977), and using $Q_W$ as the global reaction coordinate. In this method, the reaction coordinate range is divided in a number of consecutive sampling windows, which are centred at different values of the reaction coordinate. In each window, a biasing potential is added to maintain the protein inside the window, enhancing the sampling near specific values of the reaction coordinate. By using this method, it is possible to sample at low $Q_w$ values, which would be unreachable by standard (unbiased) simulations. In this work, we divided the $Q_w$ sampling interval in 40 equally spaced windows and explored the conformational space with simulations of 10 million steps in each of the windows with a time step of 3 fs. For an intuitive understanding of its meaning, $Q_w = 0.25$ corresponds to already random looking structures, whereas $Q_w = 0.7$ corresponds to structures that typically are just a few angstroms RMSD from the native structure (Schafer *et al.*, 2014). Finally, the 40 simulations are integrated using the weighted histogram analysis method (Kumar *et al.*, 1992) to obtain the thermodynamic parameters as the Free-energy values, projected in different coordinates like $Q_w$, $Q_o$ and radius of gyration ($R_g$).

To investigate the most preferred folding routes, we also performed unbiased simulations that were carried out using the AWSEM-MD (Davtyan *et al.*, 2012), with a time step of 3 fs at the $T_F$. For the 3ANK and 4ANK, three replicas of the dynamics starting from the folded structure were performed over $3 \times 10^9$ steps.

### Energy-landscape visualisation method

ELViM uses $q_{k,l}$ to describe the distance between all pairs of conformations in the real multidimensional phase space, which is given by $\delta_{k,l} = 1 - q_{k,l}$ (Oliveira *et al.*, 2019). Starting with an ensemble of structures obtained from the simulations, ELViM applies this metric for each pair of conformations, in order to build a dissimilarity matrix $\bar{\bar{\delta}}$. Next, these data may also be processed so that sufficiently similar conformations (classified by the metric) are clustered into a single representative conformation. The goal in this step is to decrease the number of conformations to be visualised. Finally, a multidimensional projection is obtained that represents each conformation in a reduced 2D effective phase space, obtaining distances given by a new distance matrix $d_{k,l}$. The method aims to minimise $|\delta_{k,l} - d_{k,l}|$, for all pairs of conformations $k$, $l$. In other words, the computation of the optimal projection consists of a minimisation procedure in which all the distances between clusters in the real multidimensional phase space $\delta_{k,l}$ are made to correspond to equal distances $d_{k,l}$ in a 2D phase space. This type of method is generally known as a multidimensional scaling method (Cox and Cox, 2000; France and Carroll, 2011). In this analysis, it is possible to visualise and identify the routes and transition states between free-energy barriers, mapping the trajectories without the need for a reference conformation or reaction coordinates.

### Results and discussion

#### 3ANK: The simplest and well-behaved funnel

The results for the 3ANK (PDB ID = 1n0q) protein, as shown in Fig. 2*a*, are consistent with a two-state folding process, with a free-energy barrier of 19.3 kT at $Q_o \approx 0.35$, with the folded state at $Q_o \approx 0.85$. In order to investigate the folding mechanism, we have analysed how the value of $Q_o$ for each residue evolves as a function of the reaction coordinate, the global $Q_o$ (Fig. 2*b*). In this figure, the ordinate is the global $Q_o$, whereas the colourmap shows the local $Q_o$, which is calculated by averaging the local $Q_o$ for each residue over the entire set of conformations with a given global $Q_o$. It is possible to see that the native interactions are not symmetrically distributed. Native contacts begin to be made by some residues in repeats R1 and R2 as the $Q_o$ approaches the transition state value of around 0.3. As the global $Q_o$ increases, the repeat R3 starts to fold at $Q_o \approx 0.6$. Fig. 2*a,b* shows that the protein undergoes an apparent two-state transition, with the nucleation starting in some regions of R2 and in parts of the N-terminal region. It should be noted, however, that the local $Q_o$ of some residues in R1 slightly decreases when the global $Q_o$ varies over the range of 0.6–0.8. Different mechanisms could be compatible with these observations, for example, multiple, parallel pathways, backtracking or kinetic traps. From the free-energy profile with respect to $Q_o$ alone, however, it is not possible to discriminate among these mechanisms.

The energy landscape of this protein analysed using ELViM results in an effective 2D phase space, as shown in Fig. 2*c*. Each point in this projection corresponds to a sampled conformation and is coloured according to its $Q_o$ value (see also Supplementary Fig. 1). It can be seen that the unfolded conformations populate the left upper region of the projection (dark blue), whereas the folded basin is made up of configurations clustered in the red right central portion of the projection (region I). According to the free-energy profile in Fig. 2*a*, transition states have a $Q_o$ value of about 0.3 and are represented by light blue dots. These transition conformations now divide up into two distinct paths, linking the folded and unfolded basins by the upper (region III) or alternatively through the lower part (region II) of the projection. We can see that there is a greater density of states in the lower part of the projection (region II), suggesting this is the more favourable folding route. For the purpose of discriminating these regions, we calculated mean contact maps for selected conformations that belong to each of these

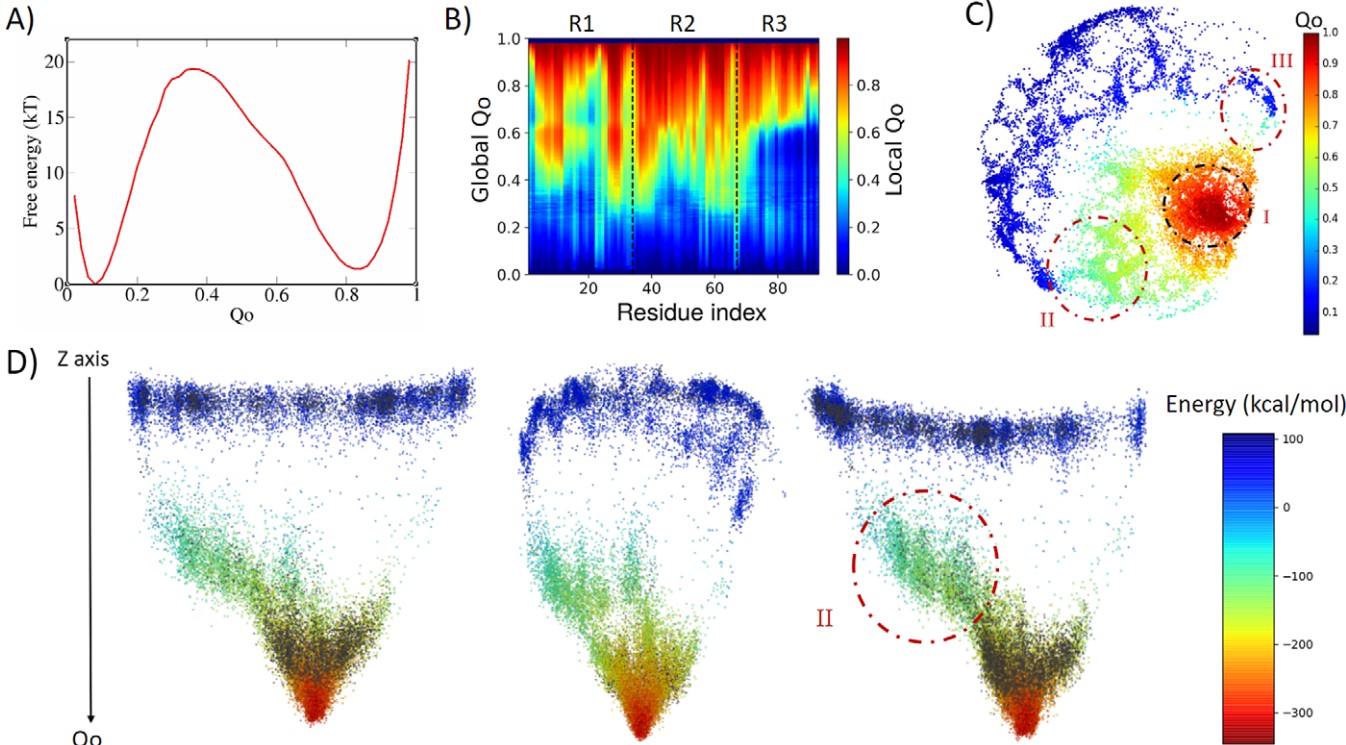

**Fig. 2.** Analysis of the biased trajectory of 3ANK. (*a*) Two-dimensional free-energy profile as a function of $Q_o$ for 3ANK. (*b*) Degree of folding of each residue as a function of $Q_o$. The colour indicates the average local $Q_o$ for a specific residue over the entire set of structures of a given global $Q_o$ for 3ANK. (*c*) Conformational phase space of 3ANK visualised through energy-landscape visualisation method (ELViM) as a function of $Q_o$. (*d*) Three-dimensional ELViM projection of the 3ANK biased trajectories in function of the dynamic total energy with the unbiased trajectory structures highlighted in black and the *Z*-axis corresponding to the $Q_o$ coordinate.

regions (available in Supplementary Figs 2 and 3). These contact maps show that region III is made up of structures in which repeats 2 and 3 are partially folded, whereas region II is composed of structures in which repeats 1 and 2 are partially folded. This analysis also reveals that these transition states have nearly the same global $Q_o$ in spite of being quite structurally different (Fig. 2*c* and Supplementary Fig. 3).

Next, we examined whether region II is indeed the kinetically more favourable folding pathway by running a set of unbiased simulations with AWSEM-MD and reconstructing the 2D projection containing structures from both the biased and the unbiased simulations (available in Supplementary Fig. 1*c*). Although the biased simulation shows a broad exploration of the structural phase space, the unbiased sampling shows a clear kinetically preferred pathway towards the native state using region II.

In addition, we have plotted a 3D representation of this effective phase space by adding a third axis, corresponding to the global $Q_o$, to the ELViM projection. In Fig. 2*d*, the colours of the dots now correspond to the total energy of each conformation, with the structures corresponding to the unbiased simulation highlighted in black, and the native state at the global minimum. This 3D projection has a funnel-like shape, and it is shown in three different views. It is possible to distinguish the main route leading to the native state, which is composed by structures of unbiased simulation and those from region II (Fig. 2*c*) clustered in its midst.

It is important to note that the conformations from region III (Fig. 2*c*) were not sampled in the unbiased simulation, suggesting that they are high-energy transition states. This result is consistent with the discussion of Cho *et al.* (2006). The fact that conformations from these two parallel routes have similar global $Q_o$ but may

nucleate starting either from the *C*- or *N*-terminal leads to the appearance of backtracking that was observed when ordering structures by its global $Q_o$ alone, as shown in Fig. 2*b*. Thus, the ELViM demonstrates that relying only on one-dimensional analysis is not always sufficient to distinguish between these two parallel sets of routes.

Lastly, we also note that there are multiple stripes of points that emerge from the global minimum, which correspond to the 'fraying' of terminal parts of repeats from the fully folded array, as previously reported by Cortajarena *et al.* (2008).

### 4ANK: Preferential path ensembles

4ANK folds in a similar way as 3ANK folds, with an apparent free-energy barrier of 19.5 kT for $Q_o \approx 0.4$, with the folded state located at $Q_o \approx 0.8$ (Fig. 3*a*). Analysing the local $Q_o$ as a function of global $Q_o$ (Fig. 3*b*), we can see that the nucleation occurs also in an asymmetric fashion, starting near the *C*-terminal repeats and propagating to the *N*-terminal repeats in a cooperative way. At the transition state ($Q_o \approx 0.3$), R3 and R4 are already folded. By the time some residues in R2 have folded ($Q_o \approx 0.4$), R4 and some residues of R3 appear to undergo an unfolding process that will be reversed again for $Q_o > 0.6$. This is very similar to what was seen for R1 in 3ANK and once more is explained by the analysis of the contact map (Supplementary Fig. 4) as an overlap of distinct families of structures with the same global $Q_o$. The 2D ELViM projection for 4ANK (Fig. 3*c*) resembles the ELViM projection for 3ANK – $Q_o$ varies smoothly from the outer left unfolded states to the central right folded basin. An important difference is that the intermediate states (shown in light blue with $Q_o \approx 0.5$) do not seem

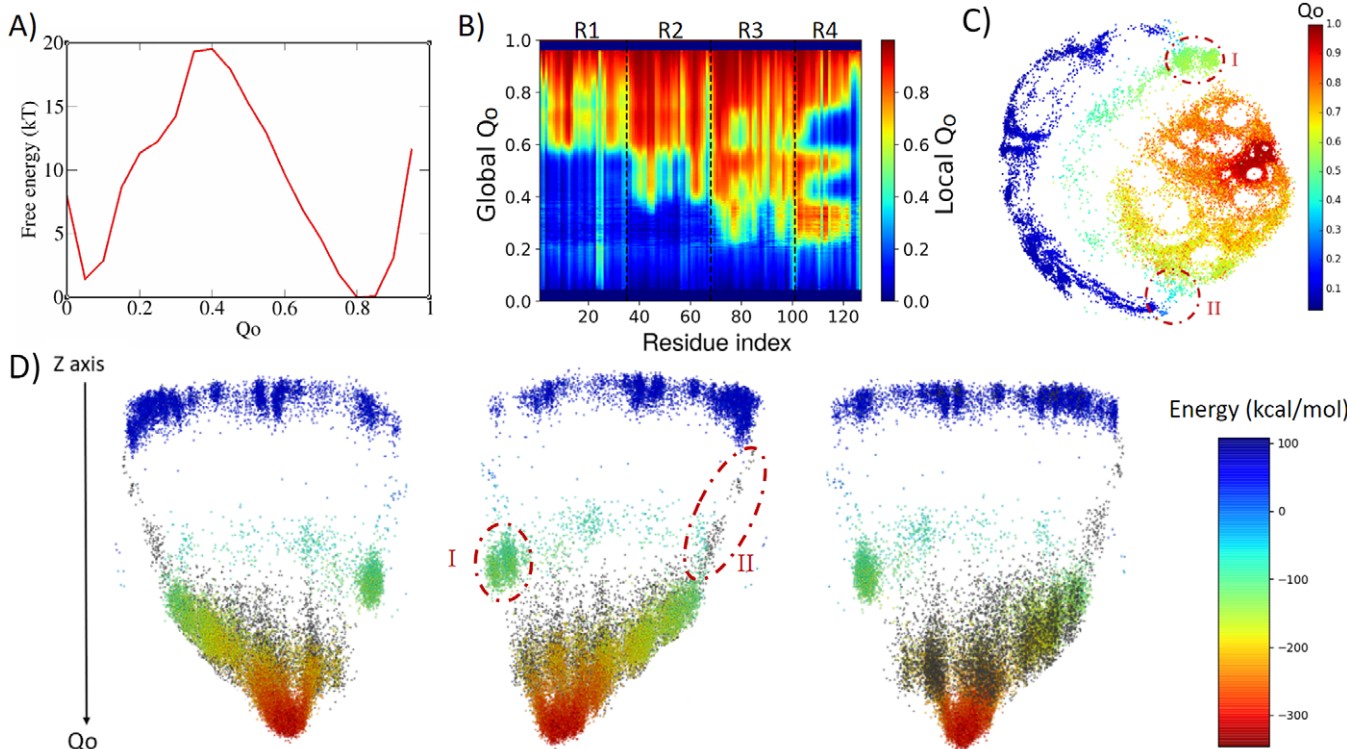

**Fig. 3.** Analysis of the biased trajectory of 4ANK. (*a*) Two-dimensional free-energy profiles as a function of $Q_o$ for 4ANK. (*b*) Degree of folding of each residue as a function of $Q_o$. The colour indicates the average local $Q_o$ for a specific residue over the entire set of structures of a given global $Q_o$ for 4ANK. (*c*) Energy-landscape visualisation method (ELViM) 2D projection of the conformational phase space of 4ANK as a function of the $Q_o$ coordinate; the transition regions I and II are indicated. (*d*) Three-dimensional ELViM projection of the 4ANK biased trajectories in function of the dynamic total energy with the unbiased trajectory structures highlighted in black and the *Z*-axis corresponding to the $Q_o$ coordinate.

to connect the folded/unfolded region in dense pathways. Instead, they form almost two separate blocks: (I) one block that is densely populated (above), and (II) another block that is densely populated below the folded basin. The mean contact maps (available in Supplementary Figs 5 and 6) indicate that region I is populated by structures which have their *N*-terminal region partially folded, whereas region II is populated by structures which have their *C*-terminal region partially folded. Thus, the ELViM energy landscape again suggests two competing parallel folding pathways. Few structures seem to join the unfolded and folded basins through the pathway I, suggesting that it represents high-energy transition structures. To test this idea, we ran a set of unbiased simulations with the AWSEM-MD and made a new 2D projection that now contains all biased and unbiased conformations (available in Supplementary Fig. 4). This new projection shows that all unbiased folding trajectories proceeded by region II, demonstrating the existence of a favourable folding pathway.

This pathway is better seen in the 3D energy surface projection shown from three different points of view in Fig. 3*d*, with the unbiased sampling represented in black. In this representation, the cluster of intermediate energy is clearly seen almost disconnected from the surface's minimum (I). To determine whether this cluster corresponds to structures that are also capable of folding, we ran several dynamic runs at different temperatures that started from structures from region I and from region II. The results show that the conformations from region II were capable of reaching the native state 76% of the time, whereas those started from region I succeeded only 37% of the time. This indicates that although the folding can occur through either region, structures in region II are more prone to reach the folded state, defining the pathway

highlighted by the unbiased trajectory as being preferred over the route with region I. Supplementary Fig. 7 shows the fraction of folding events for both regions at the different temperatures. The stripes emerging from the native basin correspond to fraying of the terminal repeats, as also seen for 3ANK.

When we try to find similarities between 3ANK and 4ANK, we observe that both indicate two possible very distinctly folding modes. We note that the most effective folding mode for them is the same, through the initial folding of R2. From Fig. 3*b*, one may have the impression that for 4ANK, R3 is the first region to fold. However, when we see the details of ELViM projection for $Q_o < 0.5$, the folded R3 conformations occur in region I, which is in the least favourable path (see Supplementary Fig. 12). Therefore, the most likely folding path is the same, through the initial folding of R2.

### 6ANK (IκBα): The most complex with multiple folding paths

Finally, we analysed the folding of a larger array, corresponding to the IκBα protein, which consists of six ankyrin repeats. In contrast to 3ANK and 4ANK which are synthetic constructs made of identical repeats yielding highly symmetric structures, IκBα is a naturally occurring protein with a very complex energy landscape because of its larger repeat array, but also because of the presence of insertions and deletions within and between some repeats, and thus the energy distribution along the array is not symmetric. Due to 6ANK complexity and asymmetries, it presents a very distinct energy landscape when compared with the other two simpler constructs.

The free-energy profile (Fig. 4*a*) presents a much lower barrier at $Q_o \approx 0.6$ that separates the folded state at $Q_o \approx 0.8$ from an

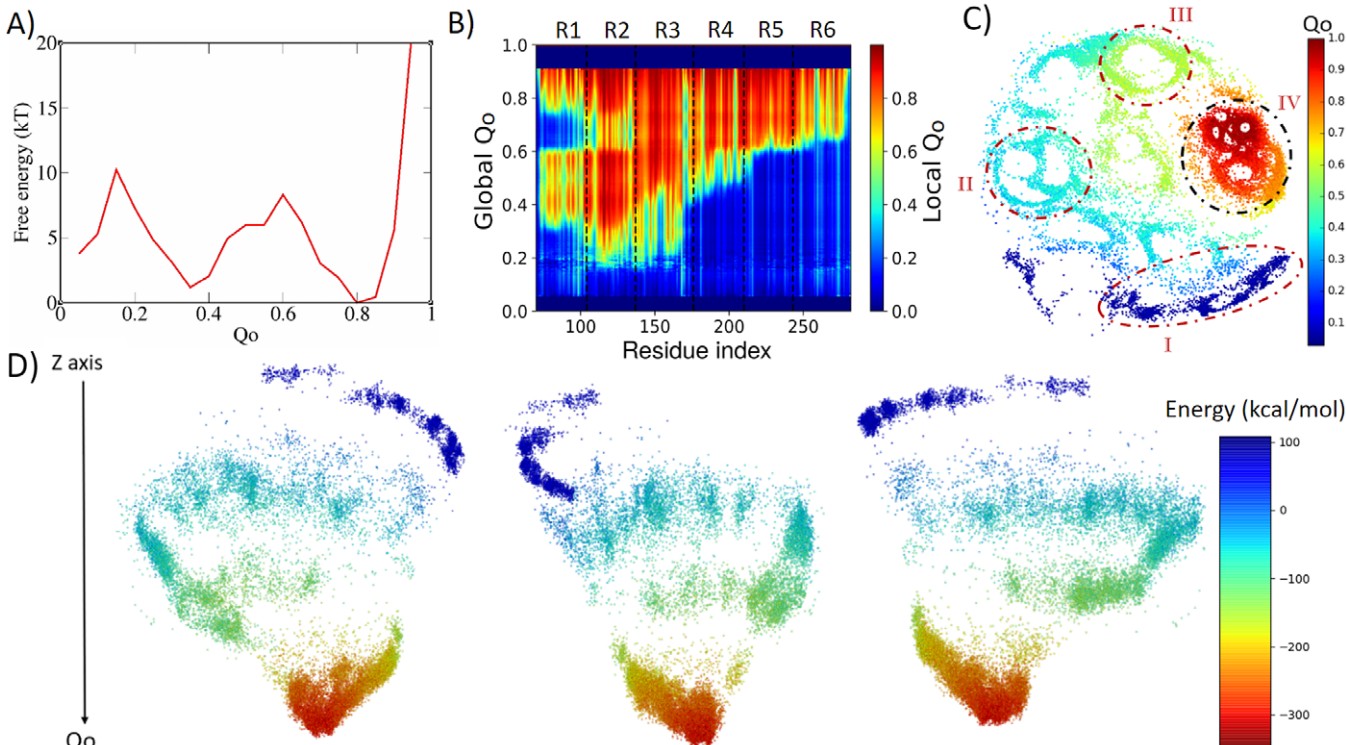

**Fig. 4.** Analysis of the biased trajectory of 6ANK. (*a*) Two-dimensional free-energy profiles as a function of $Q_o$ for 6ANK. (*b*) Degree of folding of each residue as a function of $Q_o$. The colour indicates the average local $Q_o$ for a specific residue over the entire set of structures of a given global $Q_o$ for 6ANK. (*c*) Conformational phase space of 6ANK visualised through energy-landscape visualisation method (ELViM) as a function of $Q_o$. (*d*) Three-dimensional ELViM projection of the 6ANK biased trajectories in function of the dynamic total energy, with the *Z*-axis corresponding to the $Q_o$ coordinate.

intermediary state at $Q_o \approx 0.35$, with this state corresponding to two repeats folded (R2 and R3) and four unfolded. Once again, the analysis of local $Q_o$ as a function of global $Q_o$ (Fig. 4*b*) shows that an apparent backtracking occurs in the folding of region R1 between $Q_o$ 0.6 and 0.8, that is explained by the analysis of the mean contact maps present in the support information (Supplementary Figs 8 and 9), revealing once again that the apparent backtracking is perhaps caused by there being different folding routes having structures with similar global $Q_o$.

Fig. 4*c* shows the complex landscape that drives this folding, with multiple paths with intermediate clusters leading to the folded state. It is possible to see four different regions based on the coordinate $Q_o$: (1) the dark blue (between 0 and 0.2) at the bottom, corresponding to the unfolded states; (2) the light blue (between 0.3 and 0.4) at the left, corresponding to the intermediate states; (3) the light green (around 0.5) at the upper middle, corresponding to the transition states and (4) the orange to red (0.7–1.0) at the right, corresponding to the peak of the free-energy barrier to the totally folded conformation, showing the process which the protein goes through. The complexity of the energy landscape is better illustrated by the 3D projection in Fig. 4*d*, which shows that the high $Q_o$ region is almost detached from the rest of the funnel, with few paths connecting them.

Notably, even though this protein appears highly symmetrical, the folding routes are not equally populated, revealing an asymmetry of folding. It should be noticed that the IκBα is found to be only partially folded in its free state, but it becomes fully folded once it interacts with NFκB (Ferreiro and Komives, 2010), its cognate binding partner. As in the previous cases, fraying of the terminal repeats can be visualised near the native basin.

The formation of each repeat with and without bias for the 3ANK, 4ANK and 6ANK through the ELViM are available in Supplementary Figs 10, 12, and 14 in the Supplementary Material, respectively, as a function of $Q_o$ and in Supplementary Figs 11, 13, and 15, respectively, as a function of $Q_w$.

## Conclusion

Although the overall funnel-like landscape of the AWSEM structure-based models is expected from the model construction, we see the occurrence of intermediates and transition state locations is ruled by the detailed topology of the proteins. Yet, even when the topology appears to be highly symmetrical, the population of folding routes turn out to be strongly asymmetrical. We found that all three systems appear to backtrack during the formation of some elements when monitored only using the average local $Q_o$, but this effect can be understood as arising from there being different structures from parallel folding routes that have the same global $Q_o$, as demonstrated in Supplementary Figs 11 and 13. We also note that conspicuous 'fraying' of terminal repeats arises from the fully folded conformation in the native basin, as it is expected, and has been reported for repeating arrays (Cortajarena *et al.*, 2008).

It is likely that as repeat arrays grow longer, the cooperativity between repeats breaks down, as the energy of the finite-size interfaces must get stronger such to overcome the entropy of incorporating a broken defect (Galpern *et al.*, 2020). This is observed for the largest system we analysed, IκBα that has a rather flat region of the landscape that consists of a folded subdomain of about three consecutive repeat units, which is consistent with the

structural ensemble proposed for this protein in the free unbound state and that is strongly related to its functional mechanism (Truhlar *et al.,* 2008).

**Supplementary Materials.** To view supplementary material for this article, please visit http://doi.org/10.1017/qrd.2022.4.

**Acknowledgements.** M.N.S. was supported by the Conselho Nacional de Desenvolvimento Científico e Tecnológico (CNPq; Grant 130147/2020-6). This research was supported by the Center for Theoretical Biological Physics sponsored by the NSF (Grant PHY-2019745). A.B.O. acknowledges the Robert A. Welch Postdoctoral Fellow program. P.G.W. is also supported by the D.R. Bullard-Welch Chair at Rice University (Grant C-0016). V.B.P.L. was supported by CNPq (Grant 310017/2020-3) and FAPESP (Grants 2019/22540-3 and 2018/18668-1). D.U.F. is a CONICET researcher and is supported by Grant PICT2016/1467, UBACYT, and NASA Astrobiology Institute-Enigma (Grant 80NSSC18M0093).

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
