## [Reviewer Report]

*Comments to Author*: Overall Comments:

I enjoyed this paper a lot. The authors present a novel approach to the investigation of folding pathways performed on an ideal test-case protein (multiple repeating subunits along the sequence). The work is also performed without the use of machine learning, which I appreciate in today’s computational science environment! Three different systems are studied (3ANK, 4ANK and 6ANK being 3,4 and 6 subunit repeats) using the exact same simulation procedure, in principle allowing for both independent discussion of each system’s folding behaviour and comparison between each simulation suite. With their independent system analysis, their method is able to take two structural reaction coordinates and produce a multidimensional mapped representation of the folding pathways of the protein. They are able to extract free energy barriers to folding and identify multiple folding pathways. Importantly, they are able to observe which of these pathways are dominant and the reason why (in terms of their reaction coordinates). Unfortunately, the text is currently lacking in comparison between the three different systems, which is an issue given that the addition of extra subunit repeats along the protein does not just modify the folding pathways, but appears to fundamentally change them. I therefore require extra discussions (and a few other tweaks) to be included to justify publication. Nothing major and no further simulations. Revisions are given in the order they appear in the article.

Revisions:

1. Can the authors please provide explicit detail on what σ_ij is and why it is suitable as a weighting parameter in [Disp-formula eqn1]. I appreciate that this is made clear in the references, but it is extremely important to this structural metric and has the form of a Gaussian standard deviation, so the distinction should be made explicitly clear here.

2. Can the authors please add a clarification on the umbrella sampling method for non-experts. For example "...using Qw as the global reaction coordinate to accelerate sampling at the low Qw values unreachable by standard (unbiased) simulations."

3. Each Qw value could represent a multitude of unique conformations, especially at low Qw. Can the authors please specify whether the set of conformations used represent the full conformational space available to the protein, and whether the reaction pathways / trajectories obtained are global or specific to the conformations used. (I imagine this has something to do with the ELViM multidimensional projection, but the specifics were not clear to me in the text or Reference 15).

4. On page 8 the authors claim that "Figures 2A and 2B show that the folding is highly cooperative" because different parts of the protein begin to fold at the same time. I do not believe a claim of cooperativity can be made at this stage without either explicit proof of stabilising interactions between repeats, or additional simulations of R1, R2 and R3 in isolation. I will accept, however, that cooperativity can be inferred from the folding pathways and transition states shown in the Figure 2C projection, and in the 4ANK/6ANK sections. Can the authors please relax the initial claim and instead make the claim following the discussion of transition pathways on page 9 (or explain to me in your reply why Figures 2A and 2B are sufficient).

5. In the 3ANK section, the authors find that the preferential folding pathway is when R1 and R2 fold first (in intermediate region II), implying that the N-terminal region is folded. The secondary pathway is when R2 and R3 fold first,implying that the C-terminal region is folded. In 4ANK, however, we find that the preferential intermediate region (II) is populated by partially folded C-terminal structures, and the secondary intermediate region (I) by partially folded N-terminal structures, to the extent that unbiased simulations are only observed to proceed through region II. Thus the 4ANK folding mechanism appears to be exactly opposite to that of 3ANK in terms of preferential folding, which can be seen even more clearly by comparing Figures 2B and 3B. The 6ANK system then seems to more closely resembles 3ANK than 4ANK, in that the N-terminal portion of the construct is shown to fold first in Figure 4B. Can the authors please discuss, either section by section or in the conclusions, the differences and similarities between the folding pathways of the different systems, and why they might occur.

6. In Figure 4A, we see the emergence of a second free energy barrier and thus, an intermediate (potentially stable) free energy minimum. Can the authors elucidate why this new barrier appears, and why this reduces the overall free energy barrier of 19.5kT (from the 3ANK and 4ANK systems) to just 10kT (in the 6ANK system).

7. Figure 5, the graphical abstract / contents figure, is very beautiful. I like it a lot

Minor Suggestions (just suggestions, not revisions):

1. If Figures 2A and 2B could be combined such that they share a global Qo axis that would be great. Then we could compare residue behaviour and the free energy profile directly! The same in Figures 3 and 4.

2. In Figures 2A, 2B and 2C, red implies high Qo and blue implies low Qo, whereas in Figures 2D and 2E the opposite is true, as red is used to imply high energy instead. This is visually confusing. Could you reverse the energy colour scheme? The same in Figures 3 and 4.

Typos and Grammar:

1. Pg4: in the folding of consecutive -> and in the folding of consecutive

2. Pg6: time steps -> time step

3. Pg9: In Figure 2D, the colors of the dots correspond -> In Figure 2D, the colors of the dots now correspond

4. Pg11: bellow -> below

5. Pg11: new projection show -> new projection shows

6. Pg13: The Figure 4C -> Figure 4C

7. Pg13: maybe is caused -> is perhaps caused

8. Pg15: funneldness of the -> funnel-like

---

## [Reviewer Report]

*Comments to Author*: This is an interesting paper. I recommend publication with the following changes:

1. I couldn’t find statements of what the three proteins were - their pdb files were listed but not the protein names. Is 6ANK IkBa (as mentioned towards the end of the paper)? The authors should also explain why they chose these particular proteins.

2. Some context is required to compare this new model with other models studied in the literature.

3. Broader referencing would be good. Many of the references are to reviews, and there are some papers on experimental studies missing - e.g. natural repeat proteins and the presence of intermediates and their relationship to function (Balbach), parallel folding pathways (Itzhaki).

---

## [Reviewer Report]

*Comments to Author*: Reviewer #1: This is an interesting paper. I recommend publication with the following changes:

1. I couldn’t find statements of what the three proteins were - their pdb files were listed but not the protein names. Is 6ANK IkBa (as mentioned towards the end of the paper)? The authors should also explain why they chose these particular proteins.

2. Some context is required to compare this new model with other models studied in the literature.

3. Broader referencing would be good. Many of the references are to reviews, and there are some papers on experimental studies missing - e.g. natural repeat proteins and the presence of intermediates and their relationship to function (Balbach), parallel folding pathways (Itzhaki).

Reviewer #2: Overall Comments:

I enjoyed this paper a lot. The authors present a novel approach to the investigation of folding pathways performed on an ideal test-case protein (multiple repeating subunits along the sequence). The work is also performed without the use of machine learning, which I appreciate in today’s computational science environment! Three different systems are studied (3ANK, 4ANK and 6ANK being 3,4 and 6 subunit repeats) using the exact same simulation procedure, in principle allowing for both independent discussion of each system’s folding behaviour and comparison between each simulation suite. With their independent system analysis, their method is able to take two structural reaction coordinates and produce a multidimensional mapped representation of the folding pathways of the protein. They are able to extract free energy barriers to folding and identify multiple folding pathways. Importantly, they are able to observe which of these pathways are dominant and the reason why (in terms of their reaction coordinates). Unfortunately, the text is currently lacking in comparison between the three different systems, which is an issue given that the addition of extra subunit repeats along the protein does not just modify the folding pathways, but appears to fundamentally change them. I therefore require extra discussions (and a few other tweaks) to be included to justify publication. Nothing major and no further simulations. Revisions are given in the order they appear in the article.

Revisions:

1. Can the authors please provide explicit detail on what σ_ij is and why it is suitable as a weighting parameter in Eq.1. I appreciate that this is made clear in the references, but it is extremely important to this structural metric and has the form of a Gaussian standard deviation, so the distinction should be made explicitly clear here.

2. Can the authors please add a clarification on the umbrella sampling method for non-experts. For example "...using Qw as the global reaction coordinate to accelerate sampling at the low Qw values unreachable by standard (unbiased) simulations."

3. Each Qw value could represent a multitude of unique conformations, especially at low Qw. Can the authors please specify whether the set of conformations used represent the full conformational space available to the protein, and whether the reaction pathways / trajectories obtained are global or specific to the conformations used. (I imagine this has something to do with the ELViM multidimensional projection, but the specifics were not clear to me in the text or Reference 15).

4. On page 8 the authors claim that "Figures 2A and 2B show that the folding is highly cooperative" because different parts of the protein begin to fold at the same time. I do not believe a claim of cooperativity can be made at this stage without either explicit proof of stabilising interactions between repeats, or additional simulations of R1, R2 and R3 in isolation. I will accept, however, that cooperativity can be inferred from the folding pathways and transition states shown in the Figure 2C projection, and in the 4ANK/6ANK sections. Can the authors please relax the initial claim and instead make the claim following the discussion of transition pathways on page 9 (or explain to me in your reply why Figures 2A and 2B are sufficient).

5. In the 3ANK section, the authors find that the preferential folding pathway is when R1 and R2 fold first (in intermediate region II), implying that the N-terminal region is folded. The secondary pathway is when R2 and R3 fold first,implying that the C-terminal region is folded. In 4ANK, however, we find that the preferential intermediate region (II) is populated by partially folded C-terminal structures, and the secondary intermediate region (I) by partially folded N-terminal structures, to the extent that unbiased simulations are only observed to proceed through region II. Thus the 4ANK folding mechanism appears to be exactly opposite to that of 3ANK in terms of preferential folding, which can be seen even more clearly by comparing Figures 2B and 3B. The 6ANK system then seems to more closely resembles 3ANK than 4ANK, in that the N-terminal portion of the construct is shown to fold first in Figure 4B. Can the authors please discuss, either section by section or in the conclusions, the differences and similarities between the folding pathways of the different systems, and why they might occur.

6. In Figure 4A, we see the emergence of a second free energy barrier and thus, an intermediate (potentially stable) free energy minimum. Can the authors elucidate why this new barrier appears, and why this reduces the overall free energy barrier of 19.5kT (from the 3ANK and 4ANK systems) to just 10kT (in the 6ANK system).

7. Figure 5, the graphical abstract / contents figure, is very beautiful. I like it a lot

Minor Suggestions (just suggestions, not revisions):

1. If Figures 2A and 2B could be combined such that they share a global Qo axis that would be great. Then we could compare residue behaviour and the free energy profile directly! The same in Figures 3 and 4.

2. In Figures 2A, 2B and 2C, red implies high Qo and blue implies low Qo, whereas in Figures 2D and 2E the opposite is true, as red is used to imply high energy instead. This is visually confusing. Could you reverse the energy colour scheme? The same in Figures 3 and 4.

Typos and Grammar:

1. Pg4: in the folding of consecutive -> and in the folding of consecutive

2. Pg6: time steps -> time step

3. Pg9: In Figure 2D, the colors of the dots correspond -> In Figure 2D, the colors of the dots now correspond

4. Pg11: bellow -> below

5. Pg11: new projection show -> new projection shows

6. Pg13: The Figure 4C -> Figure 4C

7. Pg13: maybe is caused -> is perhaps caused

8. Pg15: funneldness of the -> funnel-like